# Annals of Education: Teaching Climate Change and Global Public Health

**DOI:** 10.3390/ijerph21010041

**Published:** 2023-12-27

**Authors:** William N. Rom

**Affiliations:** Department of Global and Environmental Health, School of Global Public Health, New York University, 708 Broadway, New York, NY 10003, USA; william.rom@nyumc.org

**Keywords:** climate change, public health, renewable energy, heat waves, air pollution

## Abstract

The climate crisis is a health emergency: breaking temperature records every successive month, increasing mortality from hurricanes/cyclones resulting in >USD150 billion/year in damages, and mounting global loss of life from floods, droughts, and food insecurity. An entire course on climate change and global public health was envisioned, designed for students in public health, and delivered to Masters level students. The course content included the physical science behind global heating, heat waves, extreme weather disasters, arthropod-related diseases, allergies, air pollution epidemiology, melting ice and sea level rise, climate denialism, renewable energy and economics, social cost of carbon, and public policy. The methods included student engagement in presenting two air pollution epidemiological or experimental papers on fossil fuel air pollution. Second, they authored a mid-term paper on a specific topic in the climate crisis facing their locale, e.g., New York City. Third, they focused on a State, evaluating their climate change laws and their plans to harness renewable wind, solar, storage, nuclear, and geothermal energy. Students elsewhere covered regional entities’ approach to renewable energy. Fourth, the global impact was presented by student teams presenting a country’s nationally determined contribution to the Paris Climate Agreement. Over 200 Master’s students completed the course; the participation and feedback demonstrated markedly improved knowledge and evaluation of the course over time.

## 1. Introduction

Climate change due to environmental carbon pollution was highlighted as the leading global public health crisis [1]. Global warming was predicted by Svante Arrhenius more than a century ago [2]. Almost a half century ago, James Hansen alerted us to the fact that man-made carbon pollution through the burning of fossil fuels made a detectable imprint on the global average temperature [3].

Charles David Keeling developed a measurement for carbon dioxide (CO_2_) during the International Geophysical Year in 1958; he placed his instruments atop Mauna Loa representing a clean environment on the island of Hawaii [4]. His annual measurements of CO_2_ increased from ~310 ppm to ~421 ppm CO_2_ at the present time, a ~50% increase (Figure 1). The annual rate of increase was 0.5 ppm in the 1960s and was 2–3 ppm recently. At the same time, scientists working in Antarctica drilled ice cores at Dome C, reaching time periods 800,000 years ago [5,6]. Direct measurements of CO_2_ in air bubbles were never higher than 285 ppm and paralleled temperature measurements over the same time period [7]. CO_2_ represented a greenhouse gas that reflected infrared radiation in sunlight back to the surface of the Earth, similar to growing tomatoes in a winter greenhouse. Other greenhouse gases were more potent, namely methane (CH_4_) that had an 80-fold greater influence on heating than CO_2_ over two decades [8]. CO_2_ is a long-lived gas, with a third remaining in the atmosphere after a hundred years and a fifth after a millennium. Nitrous oxide and black carbon were also climate forcers, and the hydrofluorocarbons used in cooling (air conditioners and refrigerators) were over 2000 times as potent as CO_2_ [8]. Most of the massive increase in burning fossil fuels occurred over the past 50 years, increasing to 37 billion metric tons of CO_2_ pollution this past year. The United States led in cumulative emissions at 25%, followed by the European Union at 22%, China 13%, Russia 7%, and Japan 4%; currently China is the world’s largest emitter. Economic sectors for CO_2_ pollution were transportation 30%, electricity generation 28%, industry 22%, buildings 11%, and agriculture 9%. 

The global mean temperature increased to 1.3 °C from the 1880–1920 mean (Figure 2). The year 2023 shattered all global records with September 0.5 °C above the record and July and August 0.3 °C above the previous records. Over the past 15 years the rate of increase in temperature accelerated; the rate increased 40% compared to 1970–2008. Most of the heating has occurred in the northern hemisphere, on land masses, and in the Arctic that heated up four times as fast. The oceans stored most (90%) of the heat and about 25% of the CO_2_. The International Energy Agency (IEA) estimated that 80% of global energy was from fossil fuels. The remaining carbon budget was estimated at 250 Gtons to remain within 1.5 °C (50% odds, 6 years from 2023) and 1200 Gtons to remain within 2 °C (50% odds, 30 years from 2023) [9].

The Intergovernmental Panel on Climate Change (IPCC) was created in 1988 by the UN Environment Programme and World Meteorological Association to provide governments and individuals periodic assessments of scientific basis of climate change, impacts and, future risks, and options for adaptation and mitigation [10]. Hundreds of experts from 195 member countries volunteered their time to review thousands of scientific reports and develop assessments with strength of the evidence for final summaries of the recommendations based upon consensus. The IPCC stressed that the warming world was unequivocally from burning fossil fuels (anthropogenic), and that the time was running short to stay within 1.5 °C. There were three working groups (I Physical aspects of climate change, II Impacts and adaptation, and III Mitigation), with one taskforce to estimate national greenhouse gas emissions. The sixth IPCC Assessment by 743 experts was released over 2021−2023. The IPCC relied on various climate models; actual observations showed warming proceeding faster than models predicted due to externalities including El Nino events, and reduction of sulfur, nitrate, and carbon aerosols that reflected infrared radiation back into space. Reduction of aerosols in air pollution controls drove a Faustian bargain of enhanced climate heating [11]. Lastly, paleoclimate observations suggested much higher or lower sea levels with changed global temperature than being forecast by climate models. The United Nations’ Conference of 195 Parties 28 in Dubai concluded to transition away from fossil fuels equitably, triple renewable energy deployment, and reduce methane leaks in the oil and gas industries by 80–90%.

**Figure 2 ijerph-21-00041-f002:**
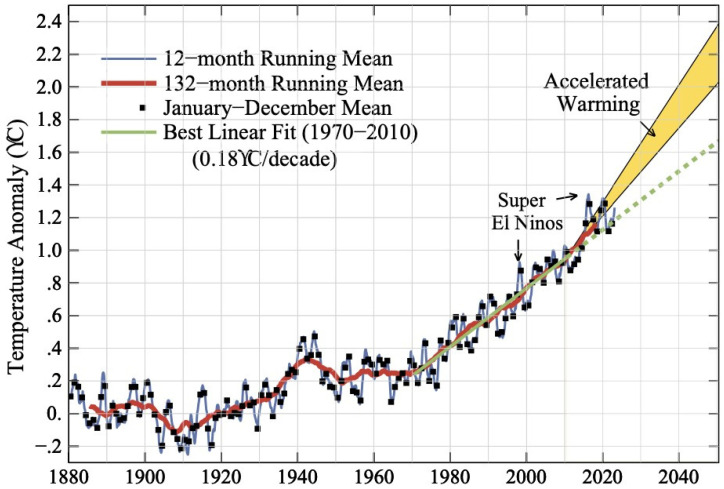
Global Temperature Relative to 1880–1920. The edges of the predicted post-2010 accelerated warming were 0.36 and 0.27 °C per decade. From Reference [11] Hansen JE, et al. with permission.

## 2. Methods

New York University School of Global Public Health had a core course in Environmental Health similar to all other accredited schools of public health. Climate change and health was covered by at least one lecture. There were multiple groups teaching and specializing in disasters that were specific to New York, e.g., World Trade Center collapse with disseminated dust, Hurricane Sandy, and the Deepwater Horizon oil spill. I noted the emergence and acceleration of the impacts of climate change and health, suggesting the need for a specific course on Climate Change and Global Public Health.

The course was designed with an introduction to the science and physics of climate change, including infrared radiation and warming of the Earth’s surface, storage in the ocean and effects on ocean circulation, the various greenhouse gases, fossil fuel combustion, global anthropogenic contributions over time, and physics of melting Greenland and Antarctic glaciers, relevant surface melt, and contribution to sea level rise. In discussions with public health students, they were surprisingly unaware of climate change and the human contribution and missed many details. 

The first assignment was to review and present two air pollution papers highlighting the epidemiological findings including background, hypothesis, study design, biostatistics, results, figures, and conclusions. Appendix B provides a list of the references 1–36 identified to represent major studies in the field of air pollution.

The second assignment was an eight-page paper on climate change affecting their locality, i.e., New York City or New York State. The students had a wide range of potential relevant topics, including 2019 Climate Leadership and Community Protection Law setting NY State targets for 40% GHG reduction from 1990 levels by 2030 and 85% reduction from 1990 levels by 2050; introduction of billion oysters for refreshing New York Harbor; congestion pricing; increasing composting; building efficiencies; gas stoves; Public Utilities Commission; increasing hydroelectric power by importations from Quebec; extending life of nuclear power plants; plans for offshore wind; plans for solar farms; heat pumps; electrifying school and public buses; increasing bike lanes; planting trees and expanding parks; hydraulic gates and dikes to protect the harbor from storm surges; heat waves; New York City Climate Laws; farmer’s markets; etc.

The third project assigned each student a State to research their wind and solar achievements and plans for renewables in the future. They presented State laws and executive orders related to climate goals or restrictions; they researched the actions of public utility commissions related to power plants, the grid, and response to federal initiatives. They presented these and other climate considerations and concluded with an assessment of the State or regional political entity’s ability to achieve reductions in CO_2_ emissions. 

The final examination involved teamwork, with 4–5 students on each team for a power-point slide presentation on a country and its Nationally Determined Contribution toward the 2015 Paris Climate Agreement. The presentations were on 4–5 countries chosen from the following list: Germany, Mexico, Canada, South Africa, India, United Kingdom, China, Denmark, Norway, Russia and Brazil. Each student presented a topic, e.g., renewables, fossil fuels, nuclear energy, transportation, agriculture and forestry, and legal goals and political hurdles.

The data collection included mid-term papers and presentations on renewable energy and countries’ plan to achieve mitigation of carbon pollution and adaptation to a warmer world. These were analyzed by the author and graded, with feedback then provided to the students. Interestingly, the students provided feedback on their colleagues’ efforts. 

## 3. Results

There were 66 Masters students on the course in the Wagner Graduate School of Public Service, followed by 137 MPH students in the NYU School of Global Public Health for a total of 203 students over 16 years. The students began knowing nil to minimal facts about climate change and global public health. Following their presentations, their knowledge base increased dramatically, and their skills improved. All demonstrated improvements in their slide and oral presentations. Two students combined and revised their mid-term papers for a manuscript for publication (covering a local law in New York City mandating building efficiencies, change in fossil fuel heating, and reduction of fracked gas). A slide presentation of California for state regulations and goals, renewables, and phase out of fossil fuels is included in the Appendix A. 

The students graded the class by the following criteria: course objectives clearly stated, well organized, intellectually stimulating, grading criteria for assignments clear, course readings and material contributed to learning, course allowed me to connect with other students, course followed expectations outlined in the syllabus, overall instructor, instructor provided an environment that was conducive to learning, helpful feedback, and allowed communication with my instructor. The evaluations from 2018 to 2020 ranged from 4.3 to 4.5 out of 5 as a perfect score, and the evaluations from 2021 to 2023 improved to 4.9 out of 5. Comments by the students were: “The professor wishes to offer every student the opportunity to become successful academically. I like that we present the air pollution studies and I enjoyed writing the midterm paper. Both assignments were a learning experience and we get the experience of practicing public speaking. Midterm paper is over a topic of our choosing. I like lots of discussion of medical crossover with climate change. Professor Rom provides a lot of necessary context to every topic, be it political or cultural”. In questioning what could be done to improve the course, the following was offered: “It could be nice to have a little bit more variety in topics as far as presentations since everyone in the class presents about air pollution for instance. Group assignments within the class should be replaced with interdepartmental collaborations”.

The course was taught as a J-Term two-week course at NYU-Shanghai. The class included a tour of a solar panel factory owned by Jinko. We had guest lecturers from the Chinese business community that covered the renewable energy industry in China. Students presented comparisons between China and the United States on Solar, Wind, Hydroelectric, Nuclear, Coal, and Oil/Gas industries. The students ranked the course a perfect 5.0.

The class lectures covered climate denialism and utilized the Yale School of Climate Communications surveys of American viewpoints over time. The ExxonMobil scientific studies from the 1970s showing that doubling CO_2_ from burning fossil fuels would increase temperature by 3 °C were presented [12]. The merchandising of doubt and statements of right-wing extremists were reviewed. The political consensus for closing the man-made ozone hole over Antarctica with the Montreal Agreement banning chlorofluorocarbons, followed by the Kigali Amendments on hydrofluorocarbons, were discussed in the context of global cooperation for solutions [13]. 

## 4. Discussion

Climate Change and Global Public Health was taught in the NYU Wagner Graduate School of Public Service to Masters students and MPH students in the NYU School of Global Public Health over 16 years. The student assignments were outlined, and evaluations were strongly positive. Importantly, the course content is summarized below for faculty to be able to implement such a course globally. 

Specific health threats from climate heating were highlighted. The World Health Organization estimated 150,000 deaths/year due to the consequences of climate change, rising to 250,000 in 2030–2050. In the business-as-usual scenario, the number of deaths escalated to 2 million/year globally, resulting in a 100 million excess mortality by 2100 [14]. The Global Burden of Disease IHME metrics listed 356,000 deaths due to excess heat in 2019 (18,750 deaths in the U.S.) [15]. By 2050, the number of extreme heat days was projected to double, directly affecting over 100 million individuals. Warm-season deaths attributable to anthropogenic climate change ranged from 37–75% [16]. In 2100, heat exposure will threaten the health of 4 billion people, according to the International Labor Organization, and productivity will decline 2.2%, resulting in USD2.4 trillion in economic losses [17]. More than 1 billion workers were exposed to high heat episodes and about a third of all exposed workers experienced negative health effects [18]. Extreme heat events claimed more lives each year in the U.S. than hurricanes, tornadoes, and lightning combined. In 2021, an alarming 46% of Americans endured at least three consecutive days of heat >100 °F, and by 2053, two-thirds of the population were projected to face perilous heat waves [15,19]. The number of days of heat >100 °F increased four-fold, which would reduce safe work time for 18 million U.S. outdoor workers, and cost >USD100 billion [20]. Black and Latino workers were disproportionately represented in low-wage high-risk jobs. Merging historical temperature data with the Health and Retirement Study, high exposure to extreme heat was associated with faster cognitive decline for Blacks and residents of poor neighborhoods, but not for whites, Hispanics, or residents of wealthier neighborhoods [21]. Heat waves disproportionately affected neonates and the elderly, e.g., in Switzerland over the last century, nearly 70% of heat-related mortality occurred in people over 80 years old. India saw a 55% rise in heat-related deaths over the past two decades, particularly in cities where many urban poor labor outdoors [22]. Heat exposure has been linked to adverse outcomes from pregnancy, including preterm birth, low birth weight, stillbirth, gestational diabetes, and preeclampsia [15]. The unusual 11–12 day extended European heat wave in 2003 showed that the elderly were susceptible due to lack of air conditioning, increasing mortality from cardiopulmonary disorders and mental conditions [23]. The urban heat island effect due to treeless, breezeless buildings and paved parking lots disproportionately affected Black Americans; they suffered from U.S. red-lining in the 1930s that restricted their opportunity for government or bank loans to upgrade their housing [24,25]. In 2021, there were 2.5 billion hours lost in the U.S. agriculture, construction, manufacturing, and service industries due to excess heat and heat waves, with USD500 billion/year lost in productivity by 2050 [26]. At 100 °F, productivity declines as much as 70%. Combined with increased humidity limiting evaporation, urban areas near the equator may become unlivable in the future [27]. 

Extreme weather events increased mortality through multiple pathways. The official death toll from Hurricane Maria (2017) in Puerto Rico was 64 deaths, whereas a stratified random sample of interviews identified more than 4600 deaths over a three-month period after the hurricane [28]. This was 70 times the official estimate of deaths, emphasizing the considerable lethality of extreme weather. One third of the deaths were attributed to delayed or interrupted health care. Climate heating caused 179 hurricanes over 32 years, resulting in 18,700 U.S. deaths, with Hurricane Katrina in 2005 leading the list with 1491 deaths [29]. Climate heating caused hurricanes to increase in intensity from greater heat stored in the ocean. Hurricanes also had a greater tendency to stall once making landfall and created deluges of rainfall [30]. Category 3 or greater hurricanes increased 8% per decade. Hurricanes in the Atlantic Ocean were twice as likely to grow from a weak storm into a major Category 3 or higher hurricane within 24 h [31]. A 1 °C increase in temperature resulted in a 7% greater ability for the atmosphere to store water vapor. Mortality occurred from injuries, infectious and parasitic diseases, cardiovascular diseases, neuropsychiatric disorders, and respiratory diseases. Over four decades, USD1.1 trillion was lost to weather disasters, and NOAA estimated that USD124 billion/year was lost over the past five years [32]. Interestingly, there were 3.3 million migrants from adverse weather, with 500,000 never returning to their damaged homes. Typhoons in the developing world increased 5-fold, with a 15% increase in intensity, causing 91% of the extreme weather deaths. In 2013, the Philippines suffered 6350 deaths from Typhoon Haiyan, which caused a USD3 billion economic loss. Heating of the ocean water by El Nino increased the intensity of hurricanes, e.g., Hurricane Otis decimated Acapulco, Mexico, on 30 October 2023, growing from category 1 to 5 within hours and unleashing winds of 165 miles per hour.

Extreme weather caused untold suffering by flooding, completely eliminating communities and pastoral lands. Satellite imaging showed that >900 floods from 2000–2018 affected 255–290 million people directly and showed growth by almost 20–24% [33]. In Pakistan in 2022, there was a severe heat wave melting many Himalayan glaciers and swelling the Indus River; monsoonal rains increased 5-fold leading to a third of the country being flooded. There were 33 million people affected, with 1700 deaths; 2 million livestock were lost as well as 200 million acres of farmland. One million houses, 22,000 schools, and 439 bridges were destroyed (total USD40 billion loss). Health effects included diarrhea, cholera, dengue, and malaria, and 5.4 million people were left without safe drinking water. Heavy rains associated with a hurricane in southeastern Europe drenched northern Libya and caused a series of dams to collapse with a muddy torrent, completely wiping out the center of the city of Derna, causing 15,000 deaths or persons missing. 

Extreme weather caused and exacerbated droughts, e.g., America’s Southwest, southern Madagascar, the horn of Africa, Syria, etc. The drought in the American Southwest reduced the Colorado River water available to 40 million residents in seven states, and left bathtub rings on the Lake Mead and Lake Powell Reservoirs. Tree ring analysis showed that almost 50% of the drought severity was anthropogenic and due to climate change [34]. The lower water level almost reached the intake pipes of the hydroelectric power turbines at Glen Canyon Dam in Utah. The Great Salt Lake shrank to its lowest level in recorded history; the dry lakebed contained heavy metals, including arsenic, that are residua from copper smelting, air pollution, and an adjacent coal-fired power plant. 

Drought and its accompanying heat reduced the harvest of rice, beans, wheat, and other plants. Elevated CO_2_ reduced protein levels by 10%, as well as B vitamins, iron, and zinc [35]. Heat increased risks of plant diseases to USD220 billion/year; insects consumed ~25% of the crop yield, and heat damaged olive groves, orange groves, cocoa, and other key foodstuffs. Drought led to civil strife when farmers in Syria migrated to cities where their demonstrations were met with violence. Heat reduced productivity and increased prices, e.g., olive oil declined 40% in Spain, Portugal, Italy, and Greece, resulting in almost a doubling of the price. Warmer and drier El Nino conditions increased childhood malnutrition up to 6 million people across the tropics, a part of the world where 20% of children were already severely underweight [36]. 

The *Aedes aegypti* mosquito expanded its range in latitude and altitude with global heating, and was the vector for dengue, yellow fever, Zika, and chikungunya [37]. There were four serotypes of dengue transmitted by the female mosquito *A. aegypti*. Interestingly, Wolbachia-infected mosquito deployments reduced flavivirus-infected *A. aegypti* by 77%, including all four serotypes of dengue [38]. There were three distinct syndromes of arboviral diseases: a febrile systemic illness, hemorrhagic fever, and encephalitis. The prevalence of vector-borne diseases has increased in recent decades, and the prevalence of malaria, dengue, West Nile virus, and Lyme disease were expected to further increase in forthcoming decades [39]. In 2020, there were 241 million cases and 627,000 deaths due to malaria, predominantly among women and children younger than 5 years in Africa. There were 390 million cases of dengue each year across more than 100 countries. Although infection provided decades of immunity against the infecting serotype, secondary infection with a different serotype increased the risk of severe disease. A tetravalent dengue vaccine based on the live attenuated DENV-2 virus that provided the genetic backbone for all four of the flaviviruses had 81% efficacy, with 95% efficacy among >20,000 study subjects against hospitalization from dengue [40]. Lyme Disease expanded in the Northeast and Midwest due to increased deer populations and exposure by humans to ticks, especially in gardening and outdoor activities. Vibrio infections of wounds were noted to increase in swimmers in the increasingly warmer Baltic Sea. *Coccidioides imitis* fungal infections increased from their incidence of 20,000 cases in the San Joaquin Valley to a far larger area due to drier and warmer conditions; a small percentage in Hispanic, Asian, or Native Americans progressed from a thin-walled pulmonary cavity to a systemic infection, requiring extensive anti-fungal treatment. 

An estimated 6.7 million persons (2019) die annually from air pollution, about half from outdoor pollution from burning fossil fuels and half from indoor air pollution due to biomass cooking [41]. The Institute for Health Metrics and Evaluation estimated that, for 2019, air pollution caused 3.55 million deaths from cardiovascular disease and 1.31 million from chronic respiratory diseases; China (1.85 million) and India (1.67 million) were most severely affected [42]. Deaths per 100,000 were also high in low-and-middle income countries like Pakistan, Nigeria, Indonesia, Bangladesh, Egypt, Russia, Ethiopia, and the Philippines. Indoor cooking with gas stoves releases particulate matter, nitrogen oxides, CO_2_, and methane, and several studies showed increases in asthma, especially in children [43,44]. Global warming increased pollen counts in various seasons across the globe [45]. Forest fires exacerbated air pollution due to particulate matter, volatile organics, and ozone. The Canadian boreal forest and coastal forests in summer 2023 experienced 5,855 fires, burning 45 million acres, and >200,000 Canadians needed evacuation. The entire capital of Yellowknife, Northwest Territories (20,000 persons) was evacuated. Air pollution alerts from smoke that descended on New York City from forest fires in Quebec occurred in June 2023. There was a 17% increase in asthma Emergency Room visits over 19 days. The risk of Quebec fires doubled due to climate change. Climate change increased the risk for extensive forest fires in Siberia in 2020 600-fold. The 2018 California Campfire killed 85 people and destroyed 18,000 homes and 120,000 acres of forest. The catastrophic firestorm in Lahaina, Maui, Hawaii, reduced the ancient capital to ashes, destroyed 3000 structures, and killed ~100 people [46]. Climate change-related forest fires burned intensely, resulting in death to animals, and left scorched earth behind with few seeds for new plants. This scorched earth caused landslides, and subsequent ash siltation of rivers killed fish and amphibians. From 2000–2019, 6.1% of PM_2.5_ and 3.6% of ozone were apportioned to landscape fire air pollution; Central Africa, Southeast Asia, South America, and Siberia experienced the highest forest fire exposures [47]. Exposures were four times higher in low-income countries, and 2.18 billion people were exposed to wildfire smoke, with an average of 9.9 days per person-year. 

Temperatures in the Arctic rose faster than the global average and ranged from an increase of 2.7 °C in the summer to 3.1 °C in the winter [48]. Higher temperatures in the Arctic summer thawed large areas of permafrost that stored 1600 Gtons of carbon; noteworthy, trees and buildings leaned and thermal karst lakes formed on the tundra. The average September extent of Arctic Sea ice declined from 7.82 million square kilometers in 1979 to 4.37 million square kilometers (1.69 square miles) in 2023, the fifth lowest in the 45-year satellite record. The multi-year ice shrank from about half of the total area to less than 5%. Open Arctic waters reduced because it absorbed heat rather than reflected sunlight. Melting Arctic Sea ice did not contribute to sea level rise, but the Arctic ecosystems were profoundly affected. Snow crabs experienced an 80% population collapse due to a warming Bering Sea. The Greenland ice sheet held enough water to raise mean global sea level by 7.4 m [49]. Greenland lost 270 billion tons of ice per year, half from surface melting and half from increased glacial dynamical imbalance [50]. Greenland’s glaciers emptied into fjords that channeled warm water toward the glacial front, increasing melting and calving. Land-based glaciers melted, especially those in the equatorial regions e.g., Mt. Kilimanjaro lost 81% of its ice [51]. The Antarctic Ice Sheet, Earth’s largest reservoir of fresh water, could, if completely melted, raise sea level by more than 60 m [52]. Western Antarctica was below sea level with ice shelves backing up large land-based glaciers [53]. The ocean had increased 1 °C below the Bellingshausen and Amundsen ice shelves, which caused ice shelf melting and cliff instability at the tongue of the Thwaites Glacier [54,55]. Marine ice shelf instability led to collapse, allowing the inland glaciers to advance; the Pine Island and Thwaites Glaciers were sliding downward under stress with increased numbers of crevasses forming [56]. There was increased glacial movement forward. Satellite measurements calculated a net mass loss of ~150 Gtons per year, compared to 1990–2010 mass loss of ~50 Gtons per year [57]. The loss of the Thwaites and Pine Island Glaciers could be equivalent to 5.3 m of sea level rise [58]. The sea ice around Antarctica reached a record low in 2023, below the 1979–2010 mean by approximately the area of Alaska. 

A 2-m rise in the ocean would inundate southern Florida, including Miami; the oceans rose about eight inches in the Anthropocene, half from melting glaciers and half from thermal expansion [59]. From 1993–2018, the seas rose 3.7 ± 0.5 mm/yr compared to 1.3 ± 0.7 mm/yr previously, and forecasts for the end of the century ranged from 6 to 30 mm/yr. The CO_2_ stored in the ocean stimulated formation of carbonic acid, lowering the pH of the ocean by 0.1 pH units. The acidity and global heating of the oceans contributed to episodes of coral bleaching, and more frequent episodes of bleaching did not allow the corals to recover leading to coral death [60]. 

Throughout the course, policy implications were discussed, and economics were discussed in the session on social cost of carbon and discount rate. A cap-and-trade bill authored by Senators McCain and Lieberman (2003-7) to reduce emissions from electricity and transportation sectors to 2004 levels by 2012, 1990 levels by 2020, and 60% below 1990 by 2050 garnered 43 votes (the author staffed Senator Hillary Clinton on the floor presentation of this bill). The State of Massachusetts sued the EPA to regulate CO_2_ under the Clean Air Act, and the U.S. Supreme Court agreed in a 5−4 decision. EPA published an Endangerment Finding in 2009 stating that greenhouse gases adversely affected human health and welfare; the EPA was required to take action to curb emissions of CO_2_, methane, nitrous oxide, hydrofluorocarbons, perfluorocarbons, and sulfur hexafluoride from vehicles, power plants, and other industries. 

International efforts to combat climate change peaked in 2015 with the Paris Climate Agreement, where the Conference of Parties voluntarily committed to reduce GHG emissions consistent with keeping global average temperature below 2 °C with a goal of 1.5 °C [61]. The United States committed to reduce GHG 26–28% below 2005 levels by 2025 and 80% by 2050. Although President Trump withdrew from the Paris Agreement, the United States immediately re-joined under President Biden. He declared that the U.S. would cut emissions by half by 2030 and reach net zero by 2050, with emissions from electricity dropping to zero in 2035. The EU pledged to cut emissions 55% from 1990 levels by 2030, and the U.K. by 68%. Over 100 countries pledged to reach net zero by 2050. Notable exceptions were China and Brazil, which pledged to peak emissions by 2030 and to reach net zero by 2060. However, details of countries’ plans showed that we were still on a path to ~2.7 °C by 2100; further tightening of fossil fuels’ emissions and expanding renewable energy were necessary. The United States and China pledged to triple global renewable energy implementation since the price for solar panels dropped 90% and wind turbines 70% over the past decade. James Hansen proposed an increasing carbon fee at the well-head with distribution to the people [62]; a carbon fee on imports from carbon-intensive manufacturing imposed by the EU, called a Carbon Border Adjustment Mechanism, assessed at the border, ensured that steel, cement, and other industrial goods would not undercut European businesses striving to meet greenhouse gas targets. 

The U.S. bipartisan Infrastructure Investment and Jobs Act (IIJA) and Inflation Reduction Act (IRA 2022) were the most important carrots to stimulate renewable energy investments via tax credits of 30% or more to build and expand wind, solar, and battery factories [63]. The IIJA provided USD39 billion to modernize mass transit and USD89.9 billion in guaranteed funding for public transit over five years. There was USD66 billion for Amtrak to modernize the northeast corridor. The IRA’s tax credits and loan guarantees amounted to a USD370 billion investment in wind, solar, nuclear, and geothermal and industries making components of clean-energy facilities over a 10-year time frame to achieve a 43–48% reduction in greenhouse gas levels on the 2005 baseline by 2035 [63]. Wind and solar were expected to grow 58 GW/year on average, twice as high as the previous record of 31 GW in 2021 [64]. Coal-generated electric power declined 38–92% from 2021 levels by 2030, and U.S. petroleum consumption declined 11–31% from 2005. Climate benefits ranged from USD44 billion to USD220 billion annually by 2030, and air pollution benefits added an additional USD9 to USD22 billion annually by 2030. Reduction of conventional air pollutants would avert 3900 premature deaths, 100,000 asthma attacks, and 417,000 lost workdays per year by 2030. The IRA confirmed Congress’s intent to treat GHGs as air pollutants for the purposes of Clean Air Act regulatory provisions [64].

Electric vehicles received a USD7500 tax credit depending on the sales price and origin of materials (U.S. or free-trade partners Chile, Canada, Australia, Mexico). EPA draft rules for automobiles 2027–2032 proposed a reduction of tailpipe pollutants by half; EVs would be 67% of new car sales by 2032, thereby reducing carbon emissions by ~10 billion tons over three decades. The IIJA funded USD7.5 billion to build 500,000 fast charging stations every 50 miles along 75,000 Interstate and U.S. highways. 

EPA proposed regulation of fossil-fuel power plants to reduce their carbon footprint by 90% after 2040. The U.S. reduced its coal-generated electricity by half (1.7 trillion kWh to 828 billion kWh) over a decade, but natural gas increased. The new Carbon Standards for Fossil-Fuel Fired Power Plants to Tackle the Climate Crisis and Protect Public Health avoided 617 million metric tons of CO_2_ through 2042 and gained USD85 billion in climate and health benefits [65]. The U.S. Department of Energy received USD1.25 billion for carbon capture and sequestration (CCS) technology from the IRA; they awarded USD189 million to eight early carbon capture projects in order to evaluate front-end engineering design efforts. The U.S. DOE funded USD1.2 billion in Direct Air Capture (DAC) projects in order to remove >2 million metric tons of CO_2_ per year.

Executive orders mandated an all-of-government approach by federal agencies to reduce CO_2_ pollution. Executive orders directed federal agencies to identify and address the disproportionately adverse human health or environmental effects of their actions on minority and low-income populations, to develop a strategy for implementing environmental justice, and to promote nondiscrimination in federal programs that affected human health. The Office of Climate Change and Health Equity was created by President Biden in 2022, and Executive Orders developed the American Climate Corps to train young people for jobs in the clean energy economy [66]. The Office of Information and Regulatory Analysis promulgated a 2% Discount Rate throughout the government, making the initial cost of projects higher in order to reduce footprint and costs of carbon pollution in the future. The General Services Administration retrofitted federal buildings with heat pump technology. Upgrading the electric grid was a mainstay approach, especially interstate transport for renewable energy [67]. The IEA reported that globally, 50 million miles of power lines by 2040 would be needed for meeting Paris Climate Agreement goals for adding renewable power, switching from gasoline-powered cars to plug-in vehicles, and replacing gas furnaces with electric heat pumps. Globally, there were ~3000 GW of renewable energy waiting to connect to the grid; the current global grid needed doubling to meet demand over the next 20 years. The IIJA funded USD65 billion in clean energy transmission and grid upgrades [67]. DOE reported that the U.S. needed 47,300 gigawatt-miles of new power lines by 2035, which would expand the current grid by 57%. There was a bipartisan effort to empower the Federal Energy Regulatory Commission (FERC) to approve the routes of major electric transmission lines that pass through more than one state replicating the power the agency already had over pipelines. 

President Biden emphasized offshore wind turbines with a goal of 30 Gigawatts by 2030. The Bureau of Ocean Energy Management (BOEM) leased >800,000 acres offshore of the New York Bight between New York and New Jersey. Coastal Virginia Offshore Wind planned a 2.6 GW wind farm with 176 giant wind turbines. Vineyard Wind I utilized GE Renewable Energy’s Haliade-X wind turbine generators with 13 MW of power and a height of 248 m, 15 miles off Martha’s Vineyard. The Jones Act required vessels operating between U.S. ports be American, although almost all of the vessels capable of installing wind turbines were European. 

IRA incentives encouraged building efficiency and electrification, especially the adoption of heat pumps for space and water heating and improving efficiency of cooking stoves [43,44]. The IIJA funded USD50 billion to protect against droughts, heat, floods, and wildfires in addition to weatherization as part of the adaptation strategy to build resilience to climate change [67]. The IIJA invested USD21 billion to clean up Superfund and brownfield sites, reclaim abandoned mine land, and cap orphaned oil and gas wells. This investment benefitted communities of color since 25–30% of Black Americans and Hispanic Americans lived within three miles of a Superfund site. The Equal Justice Initiative spent 40% of federal dollars to alleviate past discriminatory practices [68]. Meeting the realities of the carbon budget required national emissions budgets and a carbon tax that protected low-income families and businesses [69]. Deforestation in the boreal forests, tropical forests of the Amazon, Congo, and Indonesia needed to be curtailed to protect carbon sinks, avoid droughts, and prevent forest fires. Healthy outcomes required adaptation, funding for renewables (solar, wind, and storage) and development of a loss and damage fund to bridge the economic differences between the global North and South [70]. Healthy climate action requires adequate finance and mandatory disclosure of corporate risk from carbon damages. Research and innovation were necessary for rapid electrification, rebuilding carbon in soil, green hydrogen, and climate communication leading to 100% engagement of the public [71]. 

## 5. Conclusions

Teaching Climate Change and Global Public Health to MPH and Masters students over sixteen years was an important achievement. The purpose of this paper was to provide a guide for global faculty to duplicate the course. Student assignments resulted in excitement for climate change and global public health topics. The course created an important professional advocacy for climate change and health within the public health community. The course is also suitable for teaching medical students. Further research is needed to enable this course to be required for graduate public health students as part of public health school accreditation. Further research is needed on a climate change curriculum including topics on energy, history of public policy on climate change, and current issues in climate change. Further research on feedback with the students is also warranted.

## Figures and Tables

**Figure 1 ijerph-21-00041-f001:**
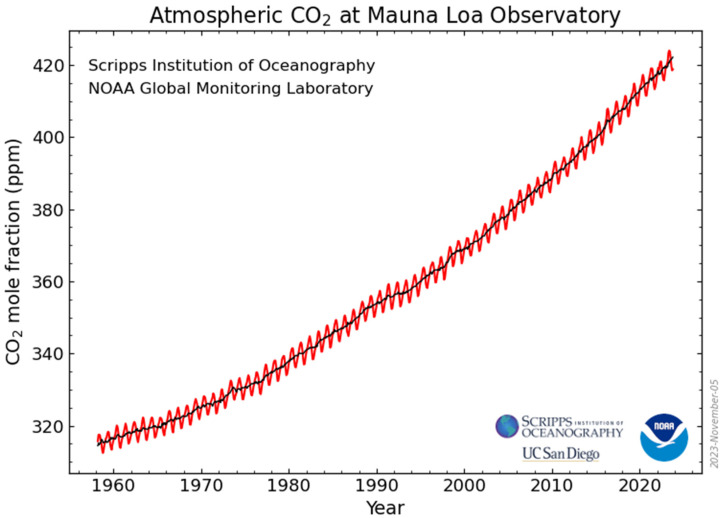
Keeling Curve. CO_2_ measurements have increased ~50% since 1958 to the present from ~310 to 421 ppm. This graph shows the full record of monthly mean carbon dioxide measured at Mauna Loa Observatory, Hawaii. The carbon dioxide data from Mauna Loa constitutes the longest record of direct measurements of CO_2_ in the atmosphere. They were started by C. David Keeling of the Scripps Institution of Oceanography in March of 1958 at the NOAA Weather Station on Mauna Loa volcano. NOAA started its own CO_2_ measurements in May of 1974, and they have run in parallel with those made by Scripps since then. (Image credit: NOAA Global Monitoring Laboratory).

## Data Availability

The data presented in this study are available on request from the author.

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
