# Peer review of "Annals of Education: Teaching Climate Change and Global Public Health"

_ijerph, 2023, doi:10.3390/ijerph21010041_

Round 1

Reviewer 1 Report

Comments and Suggestions for Authors

1. In the abstract, the authors failed to mention or describe the study methods. There should be one or two conclusions drawn from the study.

2. After giving a background to the climate crisis, the authors should discuss the efforts made in terms of teaching of climate change by previous researchers. In the concluding section of the introduction, the study's purpose and its research objectives/questions are presented.

3. The methods section needs restructuring. It must discuss scientifically, how the study was conducted. Is this an action research? Was it done qualitatively or quantitatively or mixed? How was the data from the teaching intervention collected and analyzed?  What sampling design(s) were employed in their recruitment? These matters must be discussed scholarly in the section.

There is no need to flood the section with reference articles used in the intervention. If they are indispensable in the manuscript, put them in the appendix.

4. I didn't see any scholarly results presented. The results must present data on the outcomes of the teaching intervention on climate change and global public health. The section must be set apart from the methods section. More so, it should be presented under the research objectives or questions.

5. I don't see the results on the teaching intervention discussed which is the focus of the study. The discussions are on the broad climate crisis and its associated repercussions.

6. The concluding section is superficial because it is not based on the study but offers generic conclusions on the climate crisis.

Comments on the Quality of English Language

Needs some proofreading

Author Response

I thank the Reviewers for incisive comments.  The manuscript has undergone Major Revisions as requested. First, there is a scholarly description of the purpose of the study; the Methods describes the student projects; and Results describes the assessment, evaluation, and additional presentations of the projects.  Second, the air pollution papers have been moved to an Appendix.  Third, a supplementary material has been added to illustrate a student presentation.  Fourth, two new Figures have been added on the Keeling CO2 curve and Global Average Temperature.  Fifth, the Climate Crisis data has been edited and kept providing a guide for future instructors to include in the course.  Lastly, the paper has been carefully edited for past tense and proper use of English. 

 1).  In the abstract, the authors failed to mention or describe the study methods. There should be one or two conclusions drawn from the study.

Response:  The Abstract has been re-written to describe the study methods and conclusions drawn from the study are specified.

2). After giving a background to the climate crisis, the authors should discuss the efforts made in terms of teaching of climate change by previous researchers. In the concluding section of the introduction, the study's purpose and its research objectives/questions are presented.

Response:  I agree and have discussed other researchers’ efforts at teaching environmental health and disasters with mentions of climate change.  I present the study’s purpose in Methods, Discussion, and Conclusion.

3). The methods section needs restructuring. It must discuss scientifically, how the study was conducted. Is this an action research? Was it done qualitatively or quantitatively or mixed? How was the data from the teaching intervention collected and analyzed?  What sampling design(s) were employed in their recruitment? These matters must be discussed scholarly in the section.

Response:  The Methods section now describes the approaches to student participation in the course.  There were four assignments: presentations on two air pollution papers; a mid-term paper describing a climate change project in their locale (New York State/City); assigning a State to review climate change laws and executive orders, and development of renewable energy especially wind and solar; and participation in a team presenting a country with its Paris Climate Agreement Nationally Determined Contributions. 

There is no need to flood the section with reference articles used in the intervention. If they are indispensable in the manuscript, put them in the appendix.

Response:  They have been moved to the Appendix. 

4). I didn't see any scholarly results presented. The results must present data on the outcomes of the teaching intervention on climate change and global public health. The section must be set apart from the methods section. More so, it should be presented under the research objectives or questions. 

Response:  I agree and a separate Results presents scholarly results.  These include numbers of students, an increase in climate change knowledge, an appreciation for skills in oral presentations, analysis of air pollution epidemiology, and development of student manuscripts.  An example of a State presentation is now available in Supplementary Materials.  Results of evaluations of the course are included.  A paragraph of results demonstrate that this course can be taught internationally, e.g., at NYU-Shanghai.  The Results section also includes teaching materials related to health: heat waves, extreme weather, arthropod-related diseases, and morbidity/mortality form air pollution. 

5). I don't see the results on the teaching intervention discussed which is the focus of the study. The discussions are on the broad climate crisis and its associated repercussions. 

Response:  Results now includes teaching intervention including evaluation and assessment, and topics related to climate change and health.  The evaluations were very strong, and skills attained were outstanding.  The presentations were very well done-an example of a State presentation on renewable energy is included in Supplementary Materials. 

6). The concluding section is superficial because it is not based on the study but offers generic conclusions on the climate crisis. 

Response:  The Conclusion is now based on the study as well as the importance of the course contributing to student knowledge about the climate crisis. 

Reviewer 2 Report

Comments and Suggestions for Authors

This study explored teaching climate change and global public health in campus. It has a review of climate change from different pespectives and description of teaching activities in campus, with good organization and intergration of basic knowledge on the socio-environmental impact of climate change on population. The comment are as follows, for improving the quality of the manuscript:

1. The teaching activities include four group assignments. What are the inter-relationship among the four assignments? 

2. The discussion is well explained in details. It is better to link the aspects of climate change in the discussion with the four assignments in the results part. 

3. The author's background/specialty is pulmonary medicine.  Is it possible to introduce climate change teaching to medical students, besides public health and public service students? I think it is important in helping patients reduce exposure to risk factors in clinical pratise. This may be added in the conclusion. 

Comments on the Quality of English Language

The manuscript needs a proofreading before being accepted for publication. 

Author Response

I thank the Reviewers for incisive comments.  The manuscript has undergone Major Revisions as requested. First, there is a scholarly description of the purpose of the study; the Methods describes the student projects; and Results describes the assessment, evaluation, and additional presentations of the projects.  Second, the air pollution papers have been moved to an Appendix.  Third, a supplementary material has been added to illustrate a student presentation.  Fourth, two new Figures have been added on the Keeling CO2 curve and Global Average Temperature.  Fifth, the Climate Crisis data has been edited and kept providing a guide for future instructors to include in the course.  Lastly, the paper has been carefully edited for past tense and proper use of English. 

 This study explored teaching climate change and global public health in campus. It has a review of climate change from different pespectives and description of teaching activities in campus, with good organization and intergration of basic knowledge on the socio-environmental impact of climate change on population. The comment are as follows, for improving the quality of the manuscript:

1). The teaching activities include four group assignments. What are the inter-relationship among the four assignments? 

Response:  They are linked by health aspects of climate change and solutions.  The paper describes a health aspect in New York City/State, and the air pollution papers ascribe the major health outcomes to fossil fuels.  The latter two assignments are on solutions highlighting States’ approach to renewable energy, and the final is on one of four countries’ Nationally Determined Contributions including renewables and health.

  1. The discussion is well explained in details. It is better to link the aspects of climate change in the discussion with the four assignments in the results part. 

Response:  The revised Discussion links Methods, Results, to scholarly approaches to teaching and evaluations.  I have kept much of the policy to provide a guide to future lecturers on the subject.

  1. The author's background/specialty is pulmonary medicine.  Is it possible to introduce climate change teaching to medical students, besides public health and public service students? I think it is important in helping patients reduce exposure to risk factors in clinical pratise. This may be added in the conclusion. In the abstract, the authors failed to mention or describe the study methods. There should be one or two conclusions drawn from the study.

Response:  The author prefers to provide a roadmap for lecturers to medical students (one of my former students teaches climate change in our medical school).  Although the course is at the Master’s level, it fits perfectly well for teaching medical students.  This is now mentioned in the Conclusions

Response:  The Abstract has been re-written to describe the study methods and conclusions drawn from the study are specified.

Reviewer 3 Report

Comments and Suggestions for Authors

The paper entitled Teaching Climate Change and Global Public Health aims to draw attention upon a climate change and global public health course completed so far by 203 master course.

Even if broadly well structured and well documented on the topic of climate change and associated issues the paper rests incoherent and it cannot be published in IJERPH journal.

Both the abstract and the paper fail to clearly present the topic and the objectives of this study. We find out three research stages were developed / associated to the climate change and public health course but it is not clear when and where the study took place and the rationale for performing it (what research gaps it aimed to fill in).

Methods and results are mingled and Discussion is also a broad literature based chapter on issues associated to climate change and global health, not necessarily connected or derived from a study particularly developed in this respect.

Author Response

I thank the Reviewers for incisive comments.  The manuscript has undergone Major Revisions as requested. First, there is a scholarly description of the purpose of the study; the Methods describes the student projects; and Results describes the assessment, evaluation, and additional presentations of the projects.  Second, the air pollution papers have been moved to an Appendix.  Third, a supplementary material has been added to illustrate a student presentation.  Fourth, two new Figures have been added on the Keeling CO2 curve and Global Average Temperature.  Fifth, the Climate Crisis data has been edited and kept providing a guide for future instructors to include in the course.  Lastly, the paper has been carefully edited for past tense and proper use of English. 

The paper entitled Teaching Climate Change and Global Public Health aims to draw attention upon a climate change and global public health course completed so far by 203 master course.

1). Even if broadly well-structured and well documented on the topic of climate change and associated issues the paper rests incoherent and it cannot be published in IJERPH journal.

Response:  The Manuscript is now broadly well-structured and well documented.  The paper has had major revisions to Purpose, Methods, Results, and the Climate Crisis in a coherent manner. 

2). Both the abstract and the paper fail to clearly present the topic and the objectives of this study. We find out three research stages were developed / associated to the climate change and public health course but it is not clear when and where the study took place and the rationale for performing it (what research gaps it aimed to fill in).

Response:  I agree and a separate Results presents scholarly results.  These include numbers of students, an increase in climate change knowledge, an appreciation for skills in oral presentations, analysis of air pollution epidemiology, and development of student manuscripts.  An example of a State presentation is now available in Supplementary Materials.  Results of evaluations of the course are included.  A paragraph of results demonstrate that this course can be taught internationally, e.g., at NYU-Shanghai.  The Results section also includes teaching materials related to health: heat waves, extreme weather, arthropod-related diseases, and morbidity/mortality form air pollution. Results now includes teaching intervention including evaluation and assessment, and topics related to climate change and health.  The evaluations were very strong, and skills attained were outstanding.  The presentations were very well done-an example of a State presentation on renewable energy is included in Supplementary Materials.  The Conclusion is now based on the study as well as the importance of the course contributing to student knowledge about the climate crisis. 

 3). The AbsMethods and results are mingled and Discussion is also a broad literature based chapter on issues associated to climate change and global health, not necessarily connected or derived from a study particularly developed in this respect.

Response:  The Abstract has been re-written to describe the study methods and conclusions drawn from the study are specified. The Methods section now describes the approaches to student participation in the course.  There were four assignments: presentations on two air pollution papers; a mid-term paper describing a climate change project in their locale (New York State/City); assigning a State to review climate change laws and executive orders, and development of renewable energy especially wind and solar; and participation in a team presenting a country with its Paris Climate Agreement Nationally Determined Contributions.

Reviewer 4 Report

Comments and Suggestions for Authors

Dear author

Review: Teaching Climate Change and Global Public Health

- WELL - STRUCTURED article. 

Abstract and Introduction: they are very well and clearly written

- Very interesting article BUT: 

----------------

I see problem: lines 103-198 : 36 references  /in fact 37 references/ - I suggest to remove them to references. It is very non-standart. 

line 103: "References 1-36 are the study papers utilized for evaluation"

big problem with: in the text cited: (for example) 3 x "Pope CA" - lines: 105,112, 156 - but no in references ...

There is some system that determines the order 1-37 ? Explain please. 

 I'm confused and it can be a big problem 

--------

4. Conclusions of the study is adequate and acceptable

Author Response

I thank the Reviewers for incisive comments.  The manuscript has undergone Major Revisions as requested. First, there is a scholarly description of the purpose of the study; the Methods describes the student projects; and Results describes the assessment, evaluation, and additional presentations of the projects.  Second, the air pollution papers have been moved to an Appendix.  Third, a supplementary material has been added to illustrate a student presentation.  Fourth, two new Figures have been added on the Keeling CO2 curve and Global Average Temperature.  Fifth, the Climate Crisis data has been edited and kept providing a guide for future instructors to include in the course.  Lastly, the paper has been carefully edited for past tense and proper use of English. 

Please note that the references 1-36 in the text have been moved to an Appendix as the Reviewer suggested.

Round 2

Reviewer 1 Report

Comments and Suggestions for Authors

1. On one hand, the study seems to be narrating from a review of teaching materials and/or references and on the other hand, it is reporting on a teaching intervention. I suggest that the author limits the focus of the paper to only the teaching intervention and not on the elaboration of the content taught, discussing the content with readers in a narrative review style.

2. The entire writing needs an overhaul and thorough restructuring. While the section on the methods must discuss how the teaching intervention was carried out (this has been partly done), it must discuss the study design, data collection and analysis plan, etc. These are missing.

3. The concluding section must suggest areas of further research as well as policy and practice directions for teachers handling the teaching of climate change and global public health.

Comments on the Quality of English Language

The manuscript needs to be thoroughly proofread to fix all potential syntax and sentence structuring errors.

Author Response

The author thanks Reviewers for critique of Revision 1 and suggestions for Revision 2 of Annals of Education: Teaching Climate Change and Global Public Health.  Revision 2 includes revised syntax and structural changes throughout the manuscript.  It has been extensively re-written to meet Reviewer’s concern.  The paper has been shortened >10%.  The author has improved Methods and Results and added Future Research to the Conclusion.  The paper serves as a materials, content, and structural guide to global faculty in public health for developing a climate change and public health course.

Response Second Round to Reviewer 1.

1). 1. On one hand, the study seems to be narrating from a review of teaching materials and/or references and on the other hand, it is reporting on a teaching intervention. I suggest that the author limits the focus of the paper to only the teaching intervention and not on the elaboration of the content taught, discussing the content with readers in a narrative review style.

Response.  The author has focused the Methods and Results on the teaching intervention.  The climate change content has been moved to the Discussion.  The author has kept climate change content in Discussion to provide materials for faculty who wish to teach this course.

2). 2. The entire writing needs an overhaul and thorough restructuring. While the section on the methods must discuss how the teaching intervention was carried out (this has been partly done), it must discuss the study design, data collection and analysis plan, etc. These are missing.

Response.  The study design, data collection, and analysis plan are in Revised Methods and Results.  The Revised manuscript has been carefully edited and restructured, particularly in the Introduction and Discussion.  The potential syntax and sentence structuring errors have been corrected.

3). 3. The concluding section must suggest areas of further research as well as policy and practice directions for teachers handling the teaching of climate change and global public health.

Response.  The author has added to the Conclusion several suggestions for further research on teaching climate change and global public health.

Reviewer 3 Report

Comments and Suggestions for Authors

The article entitled Annals of Education: Teaching Climate Change and Global Public Health is an improved version of the previous paper Teaching Climate Change and Global Public Health.

Unfortunately even if certain important changes were made (e.g. mainly changes on the abstract, the creation of an appendix, the description of the assignments for students on the methods chapter, some methodological details at the beginning of results chapter), this paper does not meet quality standards to be published in IJERPH Journal.

There are serious gaps in the paper related mainly to its scientific soundness.

The reader finds out about a course on Environmental Health taught at New York University School of Global Public Health which became thanks to the initiative of the researcher a course on Climate Change and Global Public Health. In the results we find out however that there is another participating institution namely Wagner Graduate School of Public Service with a number of students in the total.

Besides a series of inadvertences like the one mentioned above the most important gap is that the scientific rationale and the aim of the study is still not very clear.

Is it about the curriculum......in relation to its improvements ...or in relation to other similar curriculums in the region…in the country...in the world? What is the innovation brought by this course ? This should be clearly and explicitly motivated and shown by the researcher.

OR IF NOT INNOVATIVE

are there any pedagogical objectives .... in relation to the number of students, to their knowledge…to the curriculum…to the environmental goals ?

How the knowledge level of students was tested/evaluated at the beginning of the course …which were the results at the end of the course ?

How the individual contribution and personal knowledge improvement was tested if the evaluated work belonged to groups of 4-5 students ? We find out that the grades improved “The evaluations from 2018 to 2020 ranged from 4.3 to 4.5 out of 5 as a perfect score, and the evaluations from 2021 to 2023 improved to 4.9 out of 5” but it is not clear for how many students, when and how. The reader of the paper finds out that in fact only “Two students achieved outstanding grades" and based "on their papers” but previously is stated that “All demonstrated improvements” were  in fact evaluated as based on “their slide and oral presentations”.

For all the above this article is unpublishable in IJERPH journal and the author is warmly encouraged to restructure and made content changes in their paper and/or possibly orient to another journal.

Author Response

The author thanks Reviewers for critique of Revision 1 and suggestions for Revision 2 of Annals of Education: Teaching Climate Change and Global Public Health.  Revision 2 includes revised syntax and structural changes throughout the manuscript.  It has been extensively re-written to meet Reviewer’s concern.  The paper has been shortened >10%.  The author has improved Methods and Results and added Future Research to the Conclusion.  The paper serves as a materials, content, and structural guide to global faculty in public health for developing a climate change and public health course.

Response to Second Round Reviewer 3.

1). Unfortunately even if certain important changes were made (e.g. mainly changes on the abstract, the creation of an appendix, the description of the assignments for students on the methods chapter, some methodological details at the beginning of results chapter), this paper does not meet quality standards to be published in IJERPH Journal.

Response.  The author very strongly disagrees with this opinion.  The Revised manuscript has improved the Abstract, moved air pollution references to an Appendix, described the assignments in the Methods, and included methodological details at the beginning of Results. This manuscript is extremely INNOVATIVE developed over 16 years of graduate teaching; it will be aspirational for universities to duplicate and implement.  The quality of the manuscript as Revised twice meets the quality standards to be published in IJERPH Journal.

2). There are serious gaps in the paper related mainly to its scientific soundness.

Response.  The author very strongly disagrees with this opinion.  The author has co-edited two scientific books on Global Climate Change and Public Health (2014) with a record number of downloads from Humana Press/Springer, and Climate Change and Global Public Health Second Edition (2021).  The Second Edition has 29 chapters written by the world’s authorities on climate change.  The Reviewer is referred to these texts for scientific soundness on climate change. 

3). The reader finds out about a course on Environmental Health taught at New York University School of Global Public Health which became thanks to the initiative of the researcher a course on Climate Change and Global Public Health. In the results we find out however that there is another participating institution namely Wagner Graduate School of Public Service with a number of students in the total.

Response.  The author taught the Climate Change and Global Public Health in the Wagner Graduate School of Public Service beginning in 2007 prior to the establishment of the NYU School of Global Public Health in 2015.  The author has taught the course solely in the NYU School of Global Public Health since its founding.

4). Besides a series of inadvertences like the one mentioned above the most important gap is that the scientific rationale and the aim of the study is still not very clear.

Is it about the curriculum......in relation to its improvements ...or in relation to other similar curriculums in the region…in the country...in the world? What is the innovation brought by this course ? This should be clearly and explicitly motivated and shown by the researcher. 

OR IF NOT INNOVATIVE

are there any pedagogical objectives .... in relation to the number of students, to their knowledge…to the curriculum…to the environmental goals ?

Response.  The Revised Manuscript makes it clear that this paper is about the curriculum on Climate Change and Global Public Health.  In this regard, the manuscript is extraordinarily INNOVATIVE.  This course highlights teaching climate change in the United States, but is relevant to teaching climate change anywhere in the world.  The wind/solar and regulations presented by students on States in the U.S., could be altered for other settings on how the political entity achieves renewable energy goals.  This is now stated in the text.

5). How the knowledge level of students was tested/evaluated at the beginning of the course …which were the results at the end of the course ?

How the individual contribution and personal knowledge improvement was tested if the evaluated work belonged to groups of 4-5 students ? We find out that the grades improved “The evaluations from 2018 to 2020 ranged from 4.3 to 4.5 out of 5 as a perfect score, and the evaluations from 2021 to 2023 improved to 4.9 out of 5” but it is not clear for how many students, when and how. The reader of the paper finds out that in fact only “Two students achieved outstanding grades" and based "on their papers” but previously is stated that “All demonstrated improvements” were  in fact evaluated as based on “their slide and oral presentations”.

Response.  The Reviewer is correct in that there was no pre-test post-test examination.  However, the students readily admitted that their knowledge of climate change was nil to minimal at the beginning of the course.  They responded very positively to the teaching assignments by the end of the course.  For example, their 8-page mid-term paper about a climate change issue in New York City or State covered fascinating subjects from the billion oysters project in New York Harbor to City Council laws on making New York’s buildings more energy efficient.  These topics are easily translated to other cities where faculty may wish to duplicate the course.  The air pollution readings were the highest quality epidemiological studies.  Presentations on their study design, exposures, and biostatistics were outstanding.  On renewable energy, the students were assigned States to cover regulatory structure and wind/solar/geothermal energy.  Presentations on renewable energy were also done at NYU/Shanghai demonstrating that this course could be taught elsewhere.  The only teams were on countries’ nationally determined contributions to the Paris Climate Agreement.  The students met on dividing the assignment, e.g., on China, students covered hydroelectric power, solar and battery storage, wind turbines, or coal-fired electric power plants.   

6). For all the above this article is unpublishable in IJERPH journal and the author is warmly encouraged to restructure and made content changes in their paper and/or possibly orient to another journal.

Response.  The author has made content changes in the paper.  It will be one of the most cited publications in IJERPH Journal.

Reviewer 4 Report

Comments and Suggestions for Authors

accepted

Author Response

The author thanks Reviewers for critique of Revision 1 and suggestions for Revision 2 of Annals of Education: Teaching Climate Change and Global Public Health.  Revision 2 includes revised syntax and structural changes throughout the manuscript.  It has been extensively re-written to meet Reviewer’s concern.  The paper has been shortened >10%.  The author has improved Methods and Results and added Future Research to the Conclusion.  The paper serves as a materials, content, and structural guide to global faculty in public health for developing a climate change and public health course.